# What Children with Neuromotor Disabilities Need to Play with Technological Games

Roberta Nossa [1,*], Matteo Porro [2], Odoardo Picciolini [2], Matteo Malosio [3], Simone Pittaccio [4], Matteo Valoriani [5], Valentina Asperti [5], Francesco Clasadonte [5], Luigi Oliveto [5], Marta Mondellini [3], Simone Luca Pizzagalli [3], Matteo Lavit Nicora [3], Jacopo Romanò [4,6], Fabio Lazzari [4,6], Lorenzo Garavaglia [4], Alessandro Scano [3], Francesca Fedeli [7], Eleonora Diella [1], Sara Meloni [2] and Emilia Biffi [1,*]

1   Scientific Institute, IRCCS Eugenio Medea, 23842 Bosisio Parini, LC, Italy; eleonora.diella@lanostrafamiglia.it
2   Fondazione IRCCS Ca' Granda Ospedale Maggiore Policlinico, Pediatric Physical Medicine & Rehabilitation Unit, 20122 Milan, MI, Italy; matteo.porro@policlinico.mi.it (M.P.); odoardo.picciolini@policlinico.mi.it (O.P.); meloni.terapia@gmail.com (S.M.)
3   Institute of Intelligent Industrial Technologies and Systems for Advanced Manufacturing, National Research Council of Italy, STIIMA—CNR, 23900 Lecco, LC, Italy; matteo.malosio@stiima.cnr.it (M.M.); marta.mondellini@stiima.cnr.it (M.M.); simone.pizzagalli@taltech.ee (S.L.P.); matteo.lavit@stiima.cnr.it (M.L.N.); Alessandro.Scano@stiima.cnr.it (A.S.)
4   Institute of Condensed Matter Chemistry and Technologies for Energy, National Research Council of Italy, ICMATE—CNR, 23900 Lecco, LC, Italy; simone.pittaccio@cnr.it (S.P.); jacopo.romano@icmate.cnr.it (J.R.); fabio.lazzari@icmate.cnr.it (F.L.); lorenzo.garavaglia@cnr.it (L.G.)
5   Fifthingenium, 20131 Milan, MI, Italy; matteo.valoriani@fifthingenium.com (M.V.); valentina.asperti@fifthingenium.com (V.A.); francesco.clasadonte@fifthingenium.com (F.C.); luigi.oliveto@fifthingenium.com (L.O.)
6   Politecnico di Milano, Department of Chemistry, Materials and Chemical Engineering "Giulio Natta" (CMIC), 20133 Milan, MI, Italy
7   FightTheStroke.org Foundation, 20125 Milan, MI, Italy; francesca@fightthestroke.org
*   Correspondence: roberta.nossa@lanostrafamiglia.it (R.N.); emilia.biffi@lanostrafamiglia.it (E.B.)

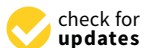



**Featured Application: The findings reported in this manuscript can act as a guideline in developing and designing videogames and consoles that are enjoyable, accessible, and inclusive for children with neurological impairments.**

**Abstract:** Game activity is fundamental for children's cognitive and social development. During recent years, technology development has led to changes in several areas, including the ludic one. However, while in the literature, there are plenty of studies that report the importance of technology-based games in rehabilitation program, little attention has been paid to their role as pure entertainment. In contrast, the market does not provide videogames that are engaging, accessible, and inclusive. In this context, a survey was distributed to families who have a child with neuromotor limitations to investigate how these children deal with play, in particular with videogames, and what the requirements are for accessible and inclusive videogames. FightTheStroke distributed the questionnaire to families with a child with neuromotor impairments in April 2020. Fifty-six families responded to the survey. The answers revealed that children generally manifest positive feelings when playing with videogames, especially with cooperative ones, even if they are not fully accessible. The survey also collected user needs and suggestions from families about the design of videogames for neuromotor-impaired children. Guidelines derived from the survey are reported for the development of entertaining, inclusive, and accessible videogames, playable by children with neuromotor disabilities.

**Keywords:** technology-based games; play; neuromotor disabilities; videogames; cerebral palsy; accessible videogames; inclusive videogames

## 1. Introduction

The 31st Article of the Convention on the Rights of the Child recognizes "the right of the child to rest and leisure, to engage in play and recreational activities appropriate to the age of the child" [1]. Indeed, gaming activities teach children to move, imagine, think, and solve problems, and they have a fundamental role in the child's cognitive and social development: through play, the child understands their inner and outer world, stimulates their abilities, and learns to accept their limits. According to Piaget, play can be classified into three types: sensorimotor, symbolic, and rule-based [2]. In particular, sensorimotor play allows children to experience their body in terms of movement; symbolic play has the function of transferring their experiences and fantasies about themselves, objects, and others, which is a fundamental experiential dimension in growth; finally, rule play has the function of releasing children from egocentricity, leading them to structure models of problem solving (indispensable for personal–social growth and maturation). Therefore, play bridges the gap between internal and external reality [3], leading the child from dependence to autonomy, and from the limitless creativity of the 5 year-old child to a system of constraints and rules, which are a fundamental prerequisite for socialization and reality understanding [4].

Notwithstanding the central importance of play in a child's life, little attention is paid to its value for children with disabilities. Approximately 5.1% of children around the world are affected by moderate or severe disabilities (e.g., motor, cognitive, vision disabilities, etc.) [5] and can present different limitations: from a deficit of coordination functions to complex situations in which movements are almost completely prevented. Therefore, these patients have specific needs and requirements and relatively few opportunities to be engaged in independent leisure activities [6].

During recent years, the panorama of available games has grown and been enhanced thanks to the development of new technologies. Nowadays, school-age children use computers, tablets, smartphones, and consoles, and the diffusion of videogames on these devices is growing steadily. As an example, we report a study of 2013 regarding children's use of mobile technologies in the United States [7]; here, the authors showed that 72% of children under 8 years old already use a mobile device for playing, compared to 38% in 2011.

In children with disabilities, technology-based play has an important role both in ludic and rehabilitative activities. Chantry et al. [8] reported that computer games can have a positive impact on disabled children. Indeed, they offer new opportunities to participate in play, facilitate autonomous and independent free play, enhance and develop children's propensity for playfulness, and enable social interaction. In addition, the interactive nature of games leads to constructive and experiential learning opportunities, promoting patients' engagement and participation in their rehabilitation programs [9]. Therefore, videogames specifically developed for a rehabilitative purpose can be considered an example of application of the gamification principle, which applies game mechanics to non-game contexts to engage audiences [10].

*Serious games* are an example of games used in rehabilitation, and they are computer/video games designed for a primary purpose other than pure entertainment. Their requirements are defined with the help of physiotherapists and clinicians in order to achieve specific rehabilitation goals. In particular, these games must respond correctly to the player's interactions, proposing challenging but not frustrating scenarios, and should include monitoring mechanisms to simplify the therapist's work [11].

Active Videogames (AVGs) are another example of videogames used for a rehabilitative purpose. AVGs are a specific category of commercial videogames that combine exercises with technology, promoting an indoor physical activity. Therefore, even though they are not specifically meant for rehabilitation, they can act as serious games for therapeutic and rehabilitative uses [12]. In the literature, there are plenty of studies that report the use of serious AVGs for rehabilitation purposes [9,12–20]. Usually, these interventions consist of sessions of commercial videogames played with commercial consoles such as

Microsoft Kinect, Sony PlayStation2 EyeToy, PlayStation Move, Nintendo Wii, Nintendo Wii Fit, GestureTech IREX, and GestureXtreme [9,12,14–18,20]. However, most of the AVGs have been specifically developed for individuals without disabilities to motivate them to increase physical activities by proposing fitness-like exercises [14]. Therefore, they could not be easily accessible in terms of understanding and playability for children with neuromotor limitations.

To overcome this situation, during recent years, some research groups have investigated the design principles of videogames to enhance [21,22] or evaluate [23] engagement during rehabilitation. Lohse et al. [21] identified several key factors in videogame design to increase motivation and engagement during rehabilitation. These key factors are:

- Reward: gameplay is motivated by rewarding experiences, regardless of the type of rewards, or the conditions that trigger rewards across games or gamers;
- Difficulty/Challenge: to avoid boredom or frustration, players should be kept at the upper limit of their ability by modulating the challenge and difficulty levels;
- Feedback: to evaluate the effectiveness of the game session, it is helpful to have feedback for improving motor learning, increasing at the same time the motivation;
- Choice/Interactivity: the player can interact with the play and choose among different options (e.g., different mini-games), increasing their connection with the virtual environment;
- Clear goals: goal-directed tasks lead to a higher chance of acceptance of assistive devices, and their lack could have a significant negative impact on the motivation of patients;
- Socialization: to increase motivation, engagement, and learning, since it allows for cooperation and competition with a partner who is a valuable source of feedback and encouragement.

The "socialization" aspect was investigated also by Pereira et al. [23], who evaluated the impact of competitive, co-active, and collaborative multi-user games on engagement and social involvement. What they found is that collaborative game modes promote more social involvement when compared to competitive and co-active modes. In accordance with Lohse, Lyons concluded that feedback, challenge, and rewards are promising mechanisms by which exergames could become more enjoyable [22].

In parallel to the identification of a suitable game design, several researchers developed videogames specifically designed for the rehabilitation of disabled individuals in order to increase the motivation of the patients or support activity participation [11,14,24–31]. These games are played on computers [24–26,28–30,32] or different commercial consoles (i.e., Nintendo Wii—Nintendo, Kyoto, Japan; Xbox 360—Microsoft, Redmond, WA, USA; PlayStation—Sony Interactive Entertainment LLC, San Mateo, CA, USA) [14,29,33–36], in some cases using commercial controllers (i.e., Nintendo Wii Balance Board, Wii Motion Plus—Nintendo, Kyoto, Japan; Microsoft Kinect—Microsoft, Redmond, United States; Novint Falcon—Novint Technologies, Albuquerque, United States; 5DT 5 Ultra Glove–Fifth Dimension Technologies, Navi Mumbai, India; EyeToy—Sony Interactive Entertainment LLC, San Mateo, CA, USA) [14,24,28–30,33–36]. On the other hand, several authors preferred to design ad hoc controllers to simplify the gaming activity or to train a specific body district. For example, Velasco et al. developed a sensorized hat to allow the users to control the cursor of the computer with movements of their heads [11]. Barton et al. designed a system based on a Stewart platform to control their custom-made computer game [26], while Betker et al. and Szturm et al. used a pressure mat [37,38]. Green and Wilson used a tabletop virtual reality-based system for the rehabilitation of children with hemiplegia [27], while Pyk et al. and Wille et al. designed a computer game controlled by data gloves in different sizes that could provide haptic feedback [39,40]. Although these rehabilitation interventions that use ad hoc videogames differ in terms of technologies applied, they are all highly configurable and based on relevant clinical activities as performed in conventional rehabilitation.

Despite the positive aspects and usefulness of technology-based therapies, videogames designed for a rehabilitative purpose are characterized by the repetition of the actions, and

they usually lack a storyline or realistic environments and characters. Therefore, they are less enjoyable and engaging than the commercial games with an entertainment function.

When referring to people (especially children) with neuromotor limitations, to date, the market is not able to meet their need for videogames that are engaging but also accessible and inclusive. To fill this gap, we investigated what disabled children need to be included in an engaging ludic activity with their peers, especially when using technology-based games, by distributing a survey in April 2020 to families who have children with neurological impairments.

Therefore, this manuscript aims to report the results of this survey, analyzing how children with neurological impairments deal with play, with particular attention to technology-based games. The results of the survey could help to define the requirements of new accessible and inclusive games with a strong technological component.

## 2. Materials and Methods

### 2.1. Survey Design and Distribution

In order to assess how children with neurological impairments (i.e., cerebral palsy (CP) and acquired brain injuries) interact with technology-based games, a survey was developed by means of REDCap, a web application that allows the management of online questionnaires and databases [41].

FightTheStroke, a foundation born in 2014 that supports young stroke survivors, children with CP, and their families, recruited families who have children with neurological impairments. The questionnaire was shared in April 2020 via a Facebook closed-group of the FightTheStroke foundation that could reach up to 1000 families who have children with neurological impairments, and parents completed it anonymously to ensure maximum privacy. The survey was administered to parents and not directly to children, since it was decided to include children with severe cognitive impairment. This allowed collecting reliable answers, also considering the impossibility of having direct contact with families and of explaining the questions to the children due to the online mode distribution of the survey forced by the COVID-19 pandemic. When possible, parents were asked to show their child pictures included in the survey in order to collect children's opinions as faithfully as possible.

The questionnaire was co-designed with families with a child with neuromotor impairments, involved by FightTheStroke, together with expert clinicians, engineers, and designers. Children were not directly involved in the survey design due to the severe cognitive impairment of some of them. However, the involvement of their families allowed designing a questionnaire suitable for them.

The survey was organized into three sections. The first one regarded child profiling in terms of pathology, type of damage, functional skills, grasping abilities, and role of technology and play in their lives; the functional skills and grasping abilities were evaluated by analyzing how children interact with daily life objects (e.g., pencil, tray, mouse, tennis ball, etc.). The second section collected information on how children play with and without technology (e.g., what does the child use to play? Does he/she play alone or with someone? Are the games competitive or cooperative?), with particular attention to the known/used consoles, the type of preferred videogames, and the feelings experienced during the game. Finally, the third section was dedicated to identifying the most suitable game design considering children with neurological impairments as final users: with the goal of collecting information on preferences, several sketches of possible game environments, characters, and control devices were proposed to the families. In addition, this section investigated how much parents would be willing to pay for inclusive videogames and consoles specifically designed for children with neurological impairments.

The survey included both open and closed-ended questions (for details, see the Supplementary Materials). Five-point Likert scales were used in the first section of the survey with two purposes:

- to evaluate the importance of the technologies in children with neurological impairments concerning specific activities (1: strongly disagree, 5: strongly agree);
- to select the difficulty level (1: no difficulties, 5: high difficulties) in doing some daily life activities (e.g., handling a pencil, a table tennis bat, etc.) in order to investigate children's gross-motor functions and their grasping abilities.

A 5 point Likert scale was also used in the second section of the survey to evaluate children's feelings during playing (both with and without technologies).

The research was approved by the institutional review board of IRCCS Medea as well as by the institute data protection officer (EU Regulation 2018/1725). Data were collected in a completely anonymous aggregated form with no personal information being collected.

### 2.2. Statistical Analysis

Median values were computed for each answer given to questions scored with the 5 point Likert scale. They are reported as median values (IQR), where IQR is the interquartile range (i.e., the difference between the 3rd and the 1st quartiles). A chi-square test was carried out on the results to verify the equidistribution of data. The Kruskal–Wallis test was run to establish whether the given answers have the same distribution for the different considered groups (i.e., children with different functional skills), while the Wilcoxon signed-ranks test was used to compare two matched samples (i.e., playing with or without the technology) and assess whether their population mean ranks differ. The statistical analysis was performed with MATLAB, and the significance was established at $p < 0.05$.

### 3. Results

As reported in the Methods section, the survey was organized into three parts: the first was dedicated to the children profiling, the second investigated how children play with and without the technology, and the third was added to identify the most suitable game design. Results are reported following this structure.

### 3.1. Section 1: Child Profiling

Fifty-six families from all over Italy and distributed as in Figure 1 responded to the survey. Parents were 42 years old on average, and the answers they gave referred to their children, who were 68% males (average age: $7 \pm 3$ years) and 32% females (average age: $6 \pm 5$ years), as shown in Figure 2.

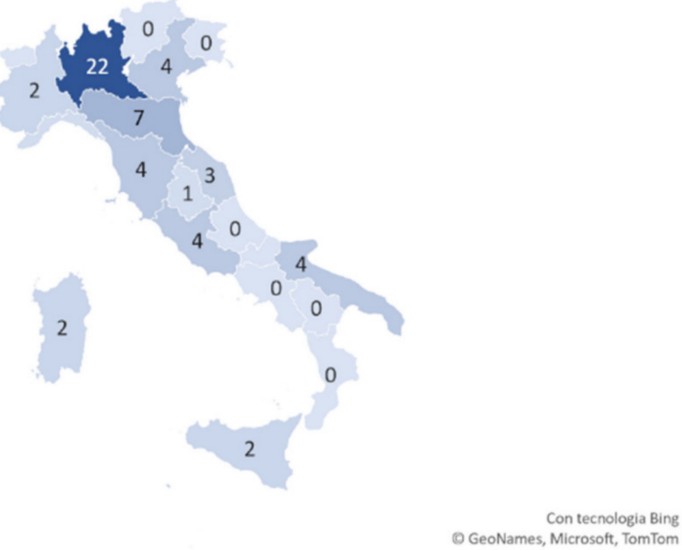

Con tecnologia Bing
© GeoNames, Microsoft, TomTom

**Figure 1.** Distribution of families that responded to the survey.

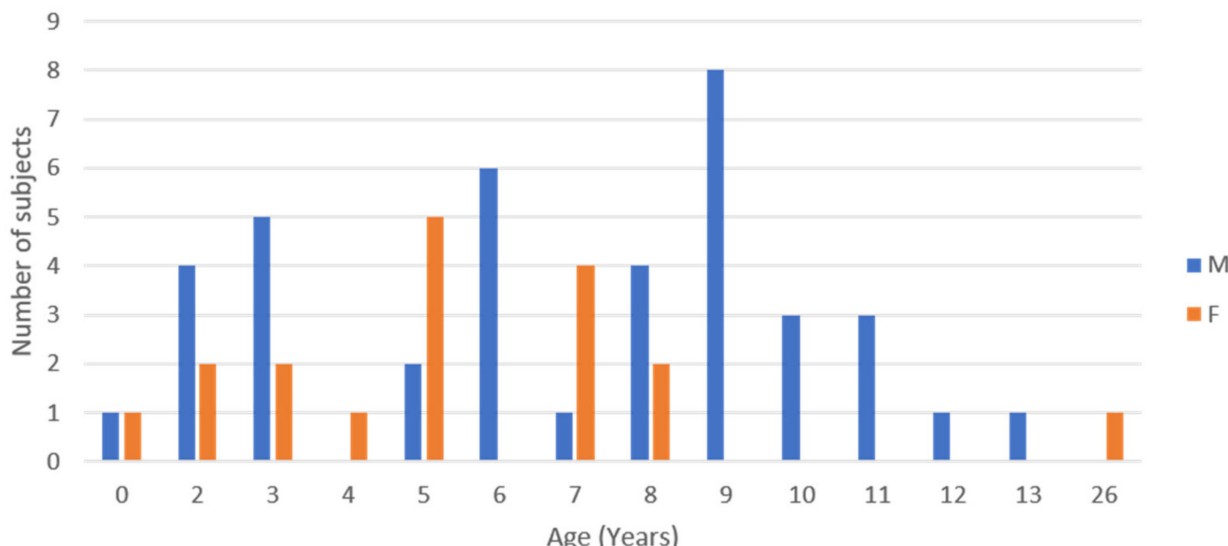

**Figure 2.** Number of children reached by the survey, divided by age. Blue bars refer to males, while orange to females.

The survey revealed that 46 children had cerebral palsy (CP), 3 acquired brain injury, while 7 families did not respond. In addition, 16 of these children also suffered from epilepsy. Among the 46 children with CP, 38 had the spastic form, 4 the dyskinetic one, 2 the ataxic one, and 2 families did not answer (Figure 3a). Considering the type of CP, 20 had the left hemiplegia, 16 the right hemiplegia, 4 the diplegia, 4 the tetraplegia, and 2 families did not respond (Figure 3b).

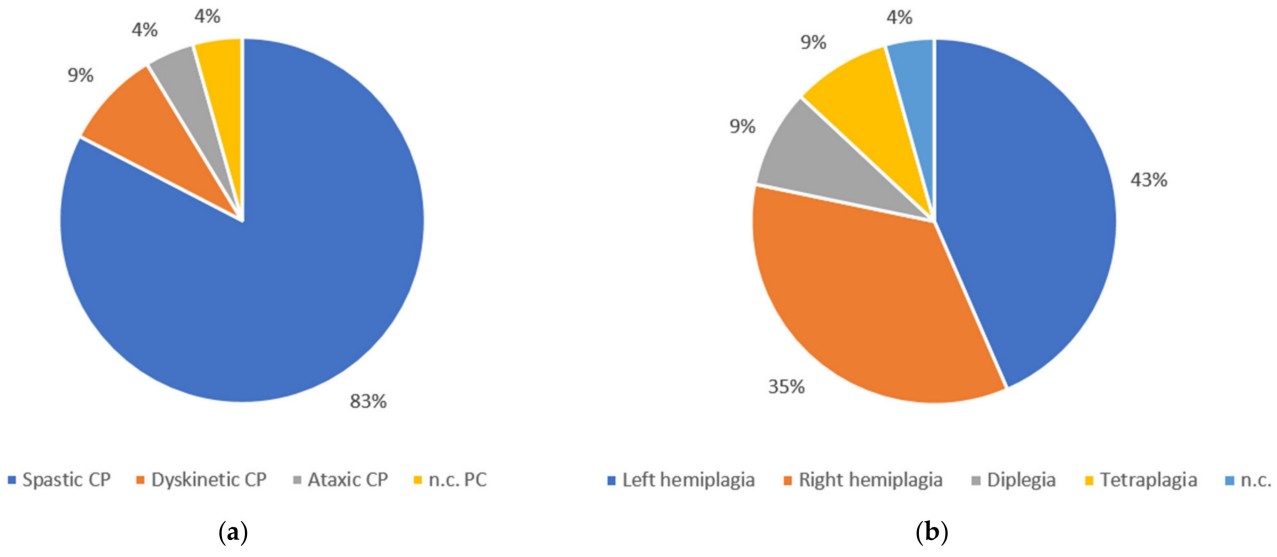

(**a**)　　　　　　　　　　　　　　　　　　　　　　　　　(**b**)

**Figure 3.** Distribution of children reached by the survey who have cerebral palsy. In (**a**), children are grouped according to the CP form, while in (**b**) according to the CP type.

The functional level of the children, measured by the gross motor function classification system (GMFCS), was mainly mild/moderate. In particular, 30 children had level I, 11 level II, 6 level III, 3 level IV, 3 level V, and 3 were not classified, as reported in Figure 4.

The cognitive functions of children were, on average, of a moderate level, with 39 children with IQs > 85, 10 with a slight delay (70 < IQs < 85), and 5 with a severe delay (IQs < 70); three were not classified, since parents did not answer.

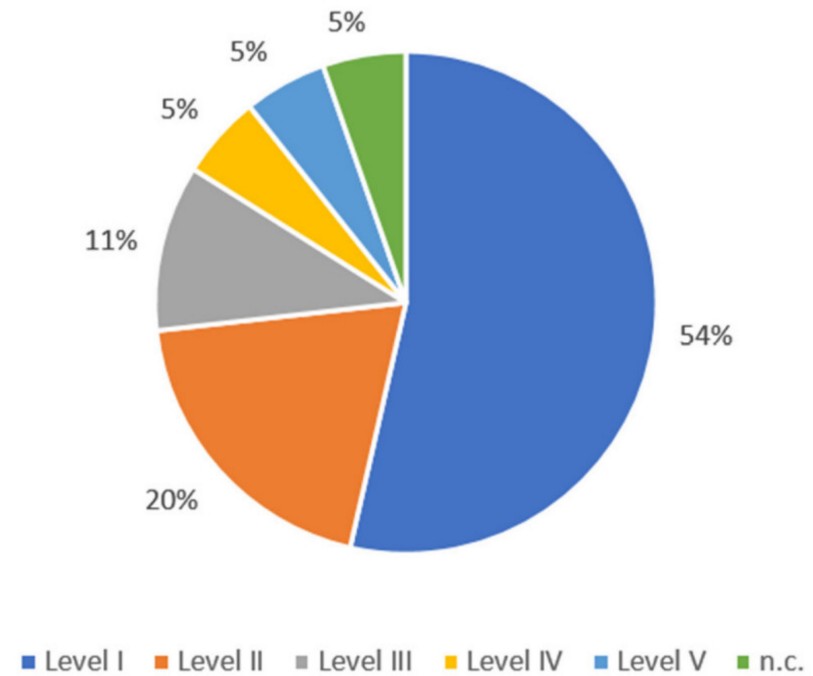

**Figure 4.** Distribution of children reached by the survey according to their GMFCS.

Subsequently, the survey investigated children's gross-motor functions and their ability to grasp cylindrical and spherical objects of different sizes in order to evaluate play accessibility with standard consoles to children with neurological impairments. The survey revealed that 52% of the subjects use only the right hand during daily actions, 36% only the left one, while 7% use both hands (three families did not respond, corresponding to 5%). In addition, 70% of the children are able to use the most impaired upper limb functionally, allowing implementing bi-manual tasks in which the less functional limb can collaborate in the game control supported by the contralateral limb. Only in a minority of cases is the most impaired limb unusable (12.5%) or an obstacle (5%) for the contralateral one (seven families did not respond, corresponding to 12.5%). The gross-motor functions were also evaluated investigating the ability to use specific objects (i.e., tablet, mouse, tray, and bicycle). The answers revealed that most of the children are able to use all the proposed objects exploiting common trunk compensation strategies (Table 1). Among those able to use the proposed objects, Table 1 also reports the percentage of who find their use easy or not, showing that most of the children easily use them.

**Table 1.** Percentage of children with neurological impairments able to use a tablet, mouse, tray, and bicycle. The percentage of those who find these objects easy or difficult to use is also specified.

| % of Children Able to Use Specific Objects | | | | | | | |
|---|---|---|---|---|---|---|---|
| **Tablet** | | **Mouse** | | **Tray** | | **Bicycle** | |
| 89% | | 64% | | 77% | | 68% | |
| **Easily** | **with Difficulties** | **without Compensation** | **with Trunk Compensation** | **with One Hand** | **Only with Both the Hands at the Same Time** | **Autonomously** | **with a Help** |
| 76% [1] | 22% [1] | 69% | 31% | 37% | 63% | 87% | 13% |

[1] The percentage sum is not 100% because a family did not answer the question.

Regarding the grasping abilities, parents answered that their child can grasp most of the objects proposed in the survey (i.e., pencil/marker, table tennis bat, table tennis ball, tennis ball) with low to medium difficulty (Table 2).

**Table 2.** Percentage of children with neurological impairments able or not in grasping a pencil/marker, table tennis bat, table tennis ball, and tennis ball. Without difficulty corresponds to score 1 in the 5-point Likert scale, high difficulty to score 5, and medium difficulty to scores 2, 3, and 4. The percentage sums are not 100% because not all the parents answered the questions.

| Tool | Graspable | Not Graspable | Graspable without Difficulty | Graspable with Medium Difficulty | Graspable with High Difficulty |
|---|---|---|---|---|---|
| Pencil/marker | 91% | 4% | 65% | 31% | 2% |
| Table tennis bat | 88% | 7% | 71% | 27% | 0% |
| Table tennis ball | 93% | 2% | 69% | 29% | 0% |
| Tennis ball | 86% | 9% | 75% | 23% | 0% |

Child profiling also included the investigation of which skills parents would like to be improved/experimented by their children through the game. With this aim, five categories were proposed: bi-manual training, precision gripping, balance improvement, gait improvement, and dynamic coordination development (e.g., jumping, stairs, height differences). Parents could select more than one option and add other skills not reported in the survey. The answers showed that 80% of them would like to practice bi-manual activities, followed by training precision gripping (71%), improving balance (68%), developing dynamic coordination (68%), and improving gait (45%). In addition, the answers to the open-ended questions highlighted the wish to train attention and concentration and to stimulate problem-solving using alternative enjoyable cognitive exercises.

Finally, since the aim of this manuscript was to analyze how children deal with technology-based games, the questionnaire evaluated the importance of technologies for children with neurological impairments in acquiring motor skills, being entertained, being included in a group, and strengthening cognitive aspects. Figure 5 analyzes the technology relevance for each activity, reporting the median score of the 5 point scale and its IQR.

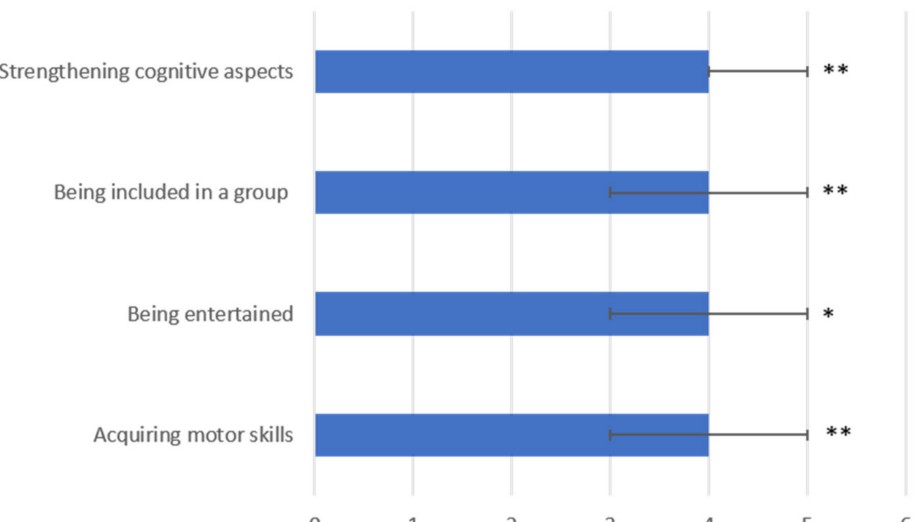

**Figure 5.** Relevance of the technology in acquiring motor skills, entertaining, being included in a group, and strengthening cognitive aspects. Chi$^2$ test: * $p < 0.05$, ** $p < 0.0001$. Null hypothesis: the scores referred to the technology relevance regarding the four proposed quality life enhancements are equidistributed with respect to the 5 point Likert scale.

Parents consider the technologies an important support for improving the level of life quality, as demonstrated by the median score always being equal to 4. Indeed, results of the chi-square test (chi$^2$ ranging between 11.9 and 41.9, $p < 0.05$) show that the distributions in Figure 5 are non-equidistributed in a statistically significant way with respect to the 5 point Likert scale but have a prevalence of positive scores ($\geq 3$). In addition, the results of the Kruskal–Wallis test, which was performed to establish whether the technology

relevance has the same distribution for the different considered groups (i.e., children with different CP form, CP type, and GMFCS) show *p*-values always >0.05 ($p$-value$_{min}$ = 0.09, $p$-value$_{max}$ = 0.91). This means that the importance of technology in acquiring one of the proposed competences is not dependent on children's CP form, CP type, or functional level (GMFCS).

### 3.2. Section 2: How Children Play with and without the Technology

In this section, the typologies of games and activities that mostly engage children with neurological impairments were investigated, as well as if they usually play alone or with someone. The activities are summarized in Table 3, which also reports the number of preferences given by the parents.

**Table 3.** Games that mostly engage children with neurological impairments as reported by their parents.

| Game/Activity Typology | Number of Preferences |
|---|---|
| Technology-based activities (watching videos and playing videogames on smartphones, computers, tablets and consoles like Xbox, PlayStation, Wii) | 39 |
| Outdoor games (e.g., hide-and-seek, basket, football, swing, Frisbee, ball, etc.) | 22 |
| Playing with building blocks | 21 |
| Toy cars, toy trains, toy tracks, radio-controlled toy car | 14 |
| Puppets, dolls, bursters, robots | 14 |
| Board games, gaming cards, puzzles | 14 |
| Bicycle, tricycle, and scooter | 8 |
| Painting | 8 |
| Pretend play (e.g., kitchen toys, housekeeping toys, tool set station, etc.) | 9 |
| Playing with moldable materials (e.g., Pongo, Didò) | 6 |
| Role-playing games (e.g., superheroes, doctor, etc.) | 5 |
| Books and interactive books | 4 |
| Toy weapons | 3 |

The answers revealed that most of the children prefer to play with technology-based games, i.e., playing videogames on smartphones, computers, and tablets, and using consoles such as the Xbox 360, PlayStation, and Nintendo Wii. Moreover, the survey showed that they usually play with parents and siblings (55%), while the remaining 45% alone; in the case of multiplayer games, 36% of the time these are cooperative games, while 18% are competitive ones.

Subsequently, the survey investigated the role of the technology within the playful context, focusing on the videogames. The answers revealed that 46% of the children usually play videogames alone, even if 81% of them would prefer to play with friends and family. Eighty-two percent of the parents reported that their children use technological games from one to seven times a week for at least 10 min up to a maximum of 2 h. The favorite ones are those played with mobile devices (i.e., smartphone and tablet) (94%), followed by the Nintendo Wii (24%), PC games (20%), Xbox 360 (17%), PlayStation (15%), Nintendo Switch (4%), and Nintendo 3DS (2%). Among children who play with tech games, 35% of them also use console accessories that work with body movements, such as the Microsoft Kinect, Wii Nunchuck, Wii Balance Board, Wii Wheel, and Wii Motion. Moreover, 7% of the parents have bought commercial controllers adapted to the child's disability, and one family personally modified the game settings to increase the usability for their child. Parents were also asked to identify the videogame categories preferred by their children (parents could select more than one category), which were sport (42.86%), action/adventure (37.50%) and car driving games (33.93%). The number of preferences for each category is shown in Table 4.

**Table 4.** Categories of videogames preferred by children with neurological impairments as reported by their parents.

| Videogame Category | Number of Preferences | Percentage of Preferences |
|---|---|---|
| Sport games | 24 | 42.86% |
| Adventure/action games | 21 | 37.50% |
| Driving games | 19 | 33.93% |
| Simulation games | 11 | 19.64% |
| Fighting games | 8 | 14.29% |
| Shooter games | 7 | 12.50% |
| Classic games (i.e., Tetris) | 6 | 10.71% |
| Role games | 6 | 10.71% |
| Brain teaser | 4 | 7.14% |
| Other | 13 | 23.21% |

In addition, parents reported that adventure games had a quite high appreciation if organized in mini-games and associated with a goal to reach or a character/hero to identify with. In contrast, if the adventure games are too difficult to be completed or the protagonist dies, this could lead to dejection in the child.

Finally, 5 point Likert scales were used to evaluate children's feelings when playing, both with and without videogames. Therefore, the survey investigated children's entertainment, engagement, frustration, satisfaction, concentration, relaxation, restlessness, and inclusion during playing. Figure 6 shows these feelings when children play with and without videogames (blue and orange bars, respectively), reporting the median score of the 5 point scale and its IQR.

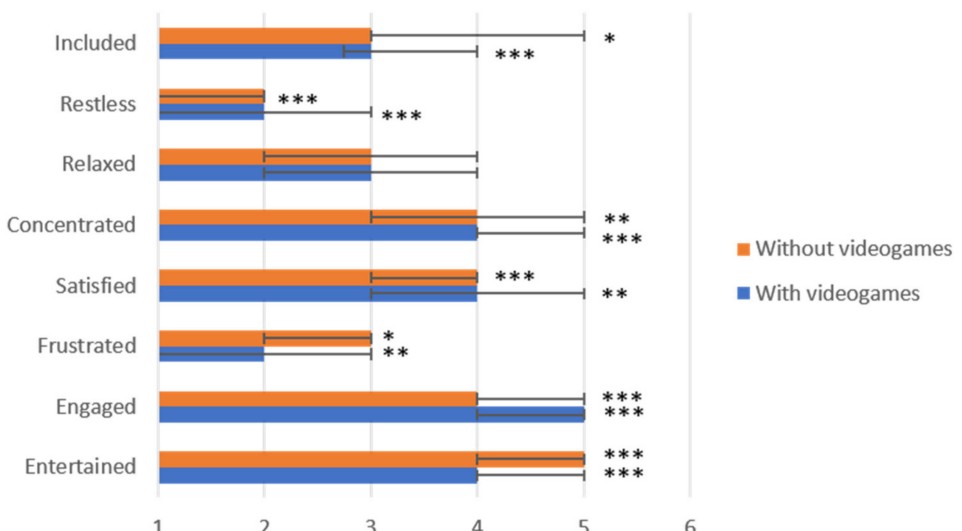

**Figure 6.** Children's feelings when playing with and without videogames (blue and orange bars, respectively). Chi$^2$ test: * $p < 0.05$, ** $p < 0.005$, *** $p < 0.0005$. Null hypothesis: the scores referred to the 8 proposed feelings that could be experienced when playing are equidistributed with respect the 5 point Likert scale.

Figure 6 shows that children generally manifest positive feelings during playing, both with and without technology, as demonstrated by the medial scores always being ≥3 if considering the positive/neutral feelings (i.e., entertainment, engagement, satisfaction, concentration, relaxation, and inclusion), and ≤3 if considering the negative ones (i.e., frustration and restlessness). Results of the chi-square test (chi$^2$ ranging between 12.4

and 65.1, $p < 0.05$) show that the score distributions in Figure 6 are non-equidistributed in a statistically significant way with respect to the 5 point Likert scale. The only exception regards the "relaxed" feeling, whereby no statistical significance was found for the games with technologies or without ($p_{w/-tech} = 0.12$, $chi^2_{w/-tech} = 7.4$, $p_{w/o-tech} = 0.13$, $chi^2_{w/o-tech} = 7.1$). Finally, the Wilcoxon signed-ranked test between the scores given to the technological games and the non-tech ones, for each of the proposed feelings, showed that there is not any statistically significant difference between the two types of play, meaning that children are not inhibited in front of technology-based games (all $p > 0.05$).

### 3.3. Section 3: Identification of the Game Design Guidelines

Section 3 aimed to evaluate the requirements that a videogame should have to be accessible and inclusive for children with neurological impairments. Questions were proposed to identify the preferred environments, characters, and console controller types, as well as the appropriate duration of a game session, which was 20/30 min. Four video game settings were proposed to the parents (sky, seabed, space, and jungle, shown in Figure 7), who had to select the most attractive for their children (more than one answer was accepted). The most successful and highly appreciated was the jungle (21 votes, Figure 7d), followed by the seabed (20 votes, Figure 7b) and the sky (18 votes, Figure 7a), while space received less approval (9 votes, Figure 7c). The answers about the favorite chromatic scale for designing the environments identified primary colors (red, yellow, green, and blue) as the winners, followed by orange, purple, and blue, with a general preference for bright and vivid colors.

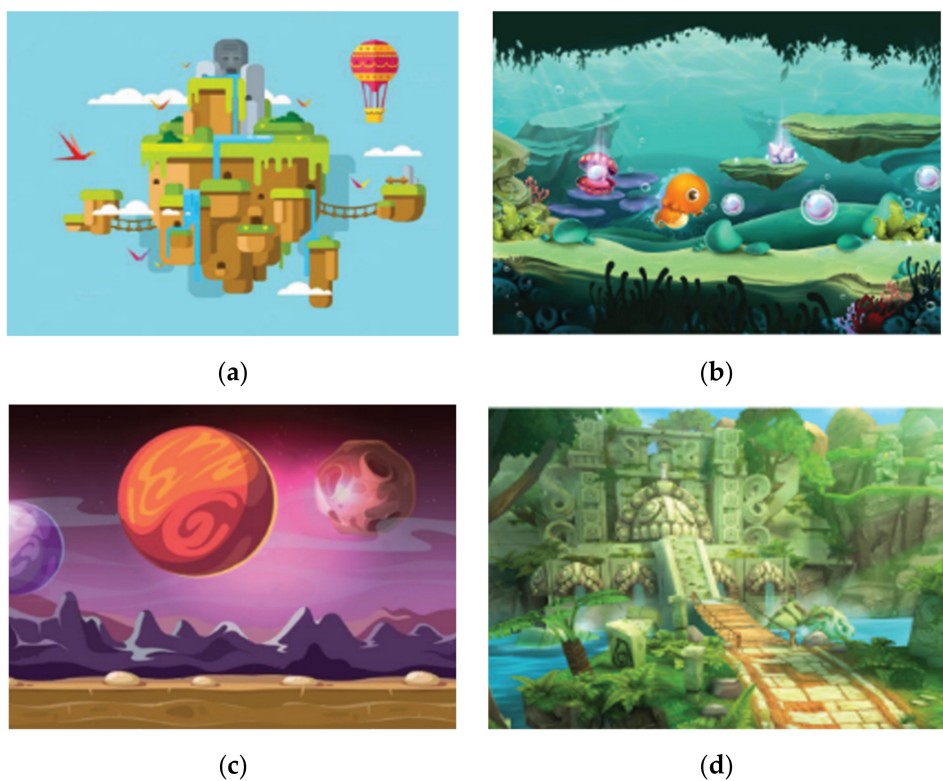

(a)　　　　　　　　　　　　　　　　　　　　　　　(b)

(c)　　　　　　　　　　　　　　　　　　　　　　　(d)

**Figure 7.** Game settings proposed to the parents among which they had to select the most attractive for their children: (**a**) the sky, (**b**) the seabed, (**c**) space, and (**d**) the jungle.

Parents were also asked to identify the most attractive character among four different options (more than one answer was accepted): the tender animal, the little monster, the little bean, and the little boy in Figure 8. The tender animal was selected as the favorite one (36 votes, Figure 8a), followed by the little boy (15 votes, Figure 8d), the little monster (11 votes, Figure 8b), and the little bean (7 votes, Figure 8c).

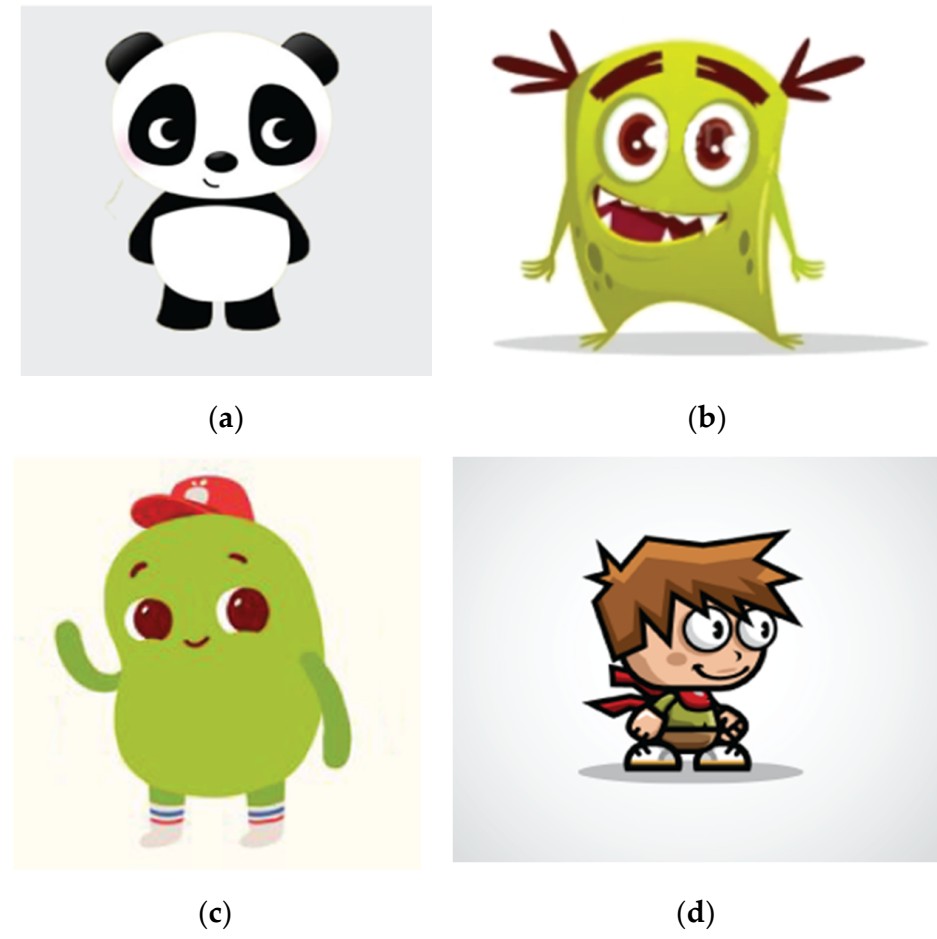

**Figure 8.** Characters proposed to the parents, from which they had to select the most attractive for their children: (**a**) the tender animal, (**b**) the little monster, (**c**) the little bean, and (**d**) the little boy.

Subsequently, parents had to select, among different game controllers (Figure 9), the most intriguing, the most enjoyable, and the one that their child would use the most.

Preferences were overlapping and similar, as shown in Table 5, which reports the five most selected devices for each of the three categories (i.e., the most intriguing, enjoyable, and used). In addition, parents' answers in Table 5 reveal that families showed a tendency to choose items they were familiar with (e.g., the steering wheel, button, etc.), while excluding the less common/known solutions.

**Table 5.** Five most selected controllers for each of the three categories (i.e., the most intriguing, enjoyable and used), as reported by parents.

| Most Intriguing Controller | | | Most Enjoyable Controller | | | Controller That Would Be Used the Most | | |
|---|---|---|---|---|---|---|---|---|
| Controller Type | Number of Preferences | % | Controller Type | Number of Preferences | % | Controller Type | Number of Preferences | % |
| Steering wheel | 27 | 12.80% | Steering wheel | 32 | 15.46% | Steering wheel | 41 | 20.50% |
| Button | 21 | 9.95% | Button | 18 | 8.70% | Button | 27 | 13.50% |
| Cloche | 15 | 7.11% | Balance board | 18 | 8.70% | Balance board | 19 | 9.50% |
| Balance board | 13 | 6.16% | Cloche | 16 | 7.73% | Cloche | 14 | 7.00% |
| Handlebar | 13 | 6.16% | Handlebar | 14 | 6.76% | Handlebar | 13 | 6.50% |

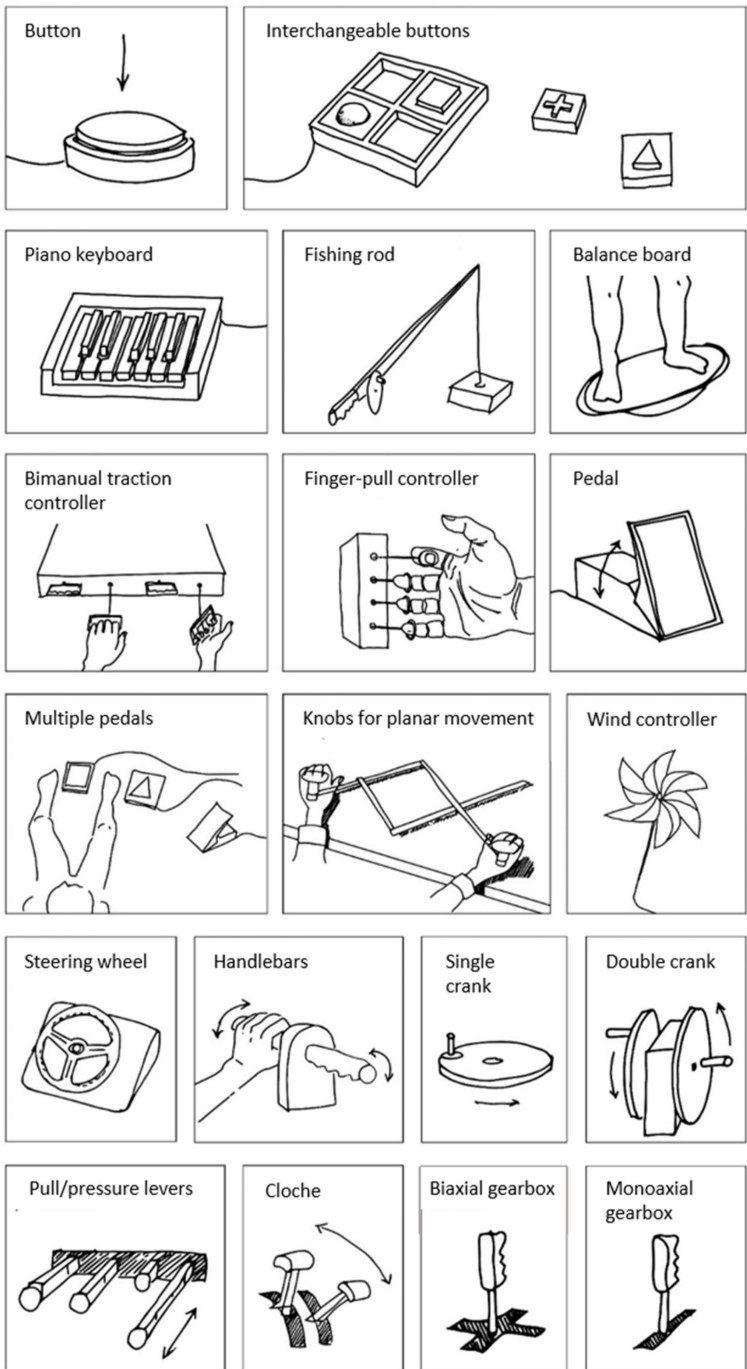

**Figure 9.** Game controllers proposed to the parents among which they had to select the most intriguing, the most enjoyable, and the one that their child would use the most.

The survey also proposed open-ended questions that investigated what parents liked, did not like, or would modify regarding the controllers that their child would use the most (third column in Table 5). Parents' answers revealed that devices that allow bi-manualisation, such as the steering wheel, the cloche, and the handlebar, were highly appreciated for two reasons: (a) with bi-manualisation, the healthy hand can support the injured one during movements (synergic use), and (b) these devices seem to facilitate grip and the development of movements and coordination of the upper limbs. In addition, parents appreciated the steering wheel and the handlebar for their affinities with daily life objects, such as the handlebar of a scooter or a bicycle and the steering wheel of a car, and, in general, for their link with symbolic play. The button was considered a simple and intuitive

device, even if monotonous and uninspiring, while the balance board was appreciated due to its usefulness in the development of balance and coordination. Parents' comments also showed there was a preference for those devices that allow the development/exercise of new skills or abilities important in everyday life and for those relatively easy to use. Indeed, the simplicity and usability of the controllers are important to avoid game abandonment due to the high frustration of the child while playing. The possibility of integrating audiovisual feedback or using different devices simultaneously seem to be desirable solutions for several parents. In general, parents gave positive feedback to the proposed controllers, even if they suggested adding some degrees of freedom for some of them (i.e., additional axial or rotational movements to the handlebar or to the cloche) and to integrate a holding support to the balance board.

Parents were also asked to indicate couples of controllers that children would like/be able to use in combination. The answers highlighted the importance of implementing multiple interactions that recall everyday actions such as driving a car, therefore combining, for example, the steering wheel with the monoaxial/biaxial gearbox, the cloche, or the car pedals. The button occurred in combination with various other devices, once also with the wind controller for the immediacy of their use in combination.

Additionally, the survey evaluated the perceived utility as a wearable game controller of a wristband, which could work as a postural stabilizer at the same time. Fifty-four percent of the parents considered the wristband useful to improve the child's posture or interaction with objects. Moreover, some free answers given to the open-ended questions explicitly support the use of the wristband as a desirable game controller. While 41% of parents suggested that it would not really help to improve the child's performance in the activities of Table 1 (three families did not respond, corresponding to 5% of the sample), the suggestions given in the open-ended questions highlight the usefulness of the wristband particularly in relation to the following activities/functions:

- Supporting the wrist and counteracting spasticity;
- Maintaining balance positions (e.g., carry a tray);
- Improving the grip of objects, especially the small ones;
- Improving the control, coordination, and strength of the upper limb;
- Improving bi-manual activities.

Finally, the survey investigated how much parents would be willing to pay for videogames and consoles with ad hoc controllers specifically designed for children with neurological impairments. Therefore, they selected, among five price categories, the one they would afford (i.e., EUR < 20, 20–40, 40–60, 60–100 and >100 for the videogames, EUR < 100, 100–250, 250–500, 500–1000 and >1000 for the consoles). The answers are summarized in Figure 10 (Figure 10a refers to the videogame, while Figure 10b to the consoles) and revealed that 55% of the parents would spend 40–100 EUR for a videogame, and 76% of them 100–500 EUR for the consoles.

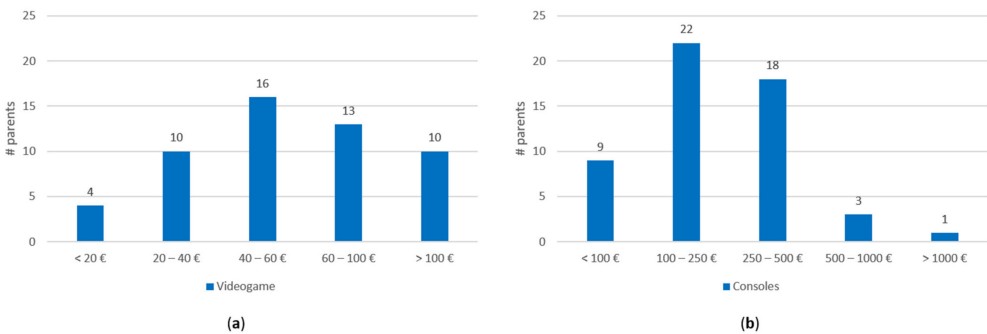

**Figure 10.** Price categories that parents would afford for videogames (**a**) and consoles (**b**) specifically designed for children with neurological impairments.

## 4. Discussion

Play has a fundamental role in children's development, and, when considering children with neurological impairments, technology-based games are an important tool both in ludic and rehabilitative activities. However, traditional technology-based games do not take into account the special needs of children with impairments, and on the other hand, games designed for disabled people are often boring for other children, thus eliminating the social component of the game. Hence, there is the necessity to develop technology-based games designed for the specific needs and characteristics of children with neurological limitations, accessible in terms of understanding and execution but also enjoyable by their peers. In order to investigate how disabled children deal with the ludic activities, in particular with technology-based games, a survey was proposed to families who have a child with neuromotor limitations. The answers showed their preferences and suggestions regarding the most suitable game design according to their child's needs.

From the survey emerged the importance of the technologies in supporting children with neuromotor problems and improving their quality of life in terms of acquiring motor skills, being entertained, being included in a group, and strengthening cognitive aspects. The answers revealed that children with neurological impairments generally manifest positive feelings when playing both with and without technology, and their favorite ludic activity consists of playing with technology-based games, highlighting the importance of developing new technologies that better meet their needs. Moreover, most of the children usually play with parents and siblings, even if there are still many that play alone and that would prefer to play with friends and family. Therefore, implementing a game able to entertain as the traditional commercial ones but, at the same time, playable by children with neuromotor limitations could improve their participation, including more players, and promote socialization.

The survey revealed that most of the children use mobile devices for playing (i.e., smartphones and tablets), followed by the Nintendo Wii, computer, Xbox 360, and PlayStation. Since both computers and consoles are widely used, the option of developing different game versions should be considered, which could be played with both of them. Despite the high appreciation of mobile games, the mobile version does not allow the integration with "smart objects" as controllers and the promotion of physical activity; therefore, it should be excluded if the ambition is to develop a videogame that encourages physical activity in addition to allowing inclusive play.

From the preferences reported by the parents, it has emerged that cooperative games are preferred over competitive ones, while the favorite game categories were sports and action/adventure, especially if organized in mini-games. In addition, the game should have a goal to reach and a character/hero to identify with to increase the engagement of the child; at the same time, the game session should not be too difficult and long to complete (20/30 min as maximum) in order to avoid the dejection in the child. The price that parents would afford for buying the inclusive videogame and console should be also taken into account. According to parents' answers, they would be willing to spend 100–250 EUR for the console and 40–60 EUR for the videogame.

The questionnaire also revealed that 35% of the children play with console controllers (i.e., Microsoft Kinect, Wii Nunchuck, Wii Balance Board, Wii Wheel, and Wii Motion), a percentage that could be increased by developing accessories customized for children with neuromotor limitations. Indeed, the controllers that present small buttons or sticks, such as the standard one of the PlayStation, are less inclusive for disabled children. Parents suggested developing accessories similar to daily life objects (e.g., a steering wheel of a car, handlebar of a scooter, etc.) that allow training bi-manual skills, so that the healthy arm/hand can support the injured one during movements; in addition, this could facilitate the development of movements and coordination, recalling daily life actions with symbolic play. On the other hand, it is possible to point out that the expressed preferences for controllers that recall daily life objects can be influenced not only by their actual usability in play, but also by the habit of using them at home or during rehabilitation. The lack

of familiarity with the other proposed controllers and the impossibility of testing the prototypes might have led to the selection of the most common/known solutions. This issue may be overcome in the future by organizing in presence testing sections, during which children and their families could test different controllers and choose the most intriguing, the most enjoyable, and the one that children would use the most. Moreover, there was a tendency to express appreciation for immediate and usable accessories such as the steering wheel or the button, while criticizing their simplicity. Therefore, it could be interesting and important to modulate the number of available interactions, proposing progressive levels of play that could follow the level of competence of the child. Finally, when designing the controllers, it is important to balance the similarity to real objects and the ease of use. Indeed, these accessories must engage the child during their use and, at the same time, avoid game abandonment due to the high frustration of the child while playing. For this reason, some parents suggested modifications to traditional controllers, such as adding a holding support to the balance board, to increase the accessibility of this object.

To our knowledge, there are no studies that investigate how children with neurological impairments deal with play, in particular with videogames. The only study that performed an investigation distributing a survey is the one proposed by Levac et al. in 2017 [42], in which authors described the clinical use of virtual reality and active videogames and identified usage barriers and facilitators. However, they reported the point of view of physical and occupational therapists, who mostly use these technologies in rehabilitation. From their survey emerged the idea that technology is perceived as useful and motivating for patients, even if barriers like the lack of funds, space, time, support staff, and appropriate clients limit the use of technologies. If Levac's manuscript is useful for making improvements regarding the use of technologies in a clinical environment, this manuscript, on the other hand, specifically focuses on the perceptions and preferences of neuromotor-impaired children and their families when playing with technologies in their home environment.

The main limit of this study is the small sample size. The reachable target population was significant, but the channel used for diffusion (Facebook) and the period of sharing (April 2020, in the middle of the COVID-19 pandemic) likely reduced the number of adhesions. In addition, almost 40% of the respondents came from one region of Italy (from Lombardy). Therefore, future studies will include more families by sharing the survey with more end-users uniformly distributed across Italy, considering the different population densities of the Italian regions (i.e., Lombardy has the highest population density). Another limitation is related to the fact that, in this manuscript, the survey collected parents' answers, but not children's ones, due to the cognitive impairment of some of them and the online administration of the survey. Therefore, future works should also collect children's opinion, administering the survey in presence in order to be able to explain questions that may be unclear to children.

## 5. Conclusions

To conclude, the findings reported in this manuscript can act as a guideline for videogame developers and console designers, since they give useful suggestions for developing technologies that are enjoyable, accessible, and inclusive at the same time. For example, these guidelines will be considered within the GiocAbile project, started in 2020, whose aim is to design a videogame for children with neurological impairments, which is as engaging as commercial ones, and understandable and playable by everyone. GiocAbile has the ambition to continuously stimulate children to play, at the same time improving their physical abilities and developing their motor skills. At a later stage of the project, GiocAbile could also be used as a rehabilitation tool with associated clinical protocols. Below, we summarize the technical requirements derived from the survey that will be also considered within the GiocAbile project:

- Avoiding designing controllers that present small buttons or sticks, since they are not easily usable by children with neuromotor impairments;

- Developing controllers that allow training the bi-manual skills, so that the healthy/less impaired arm/hand can support the injured one;
- Designing controllers that recall daily life objects (e.g., a steering wheel of a car, handlebar of a scooter, etc.), since they are intuitive to use and facilitate disabled children;
- When designing the controllers, balancing the similarity to real objects and ease of use in order to avoid frustration in children;
- Designing controllers that allow training functional movements (e.g., driving a car, carrying a tray, fishing);
- Considering developing games that can be played both with computers and consoles;
- Preferring to develop cooperative games rather than competitive ones, since they are more inclusive;
- Preferring to develop sports and action/adventure videogames, especially if organized in mini-games, with a goal to reach and a character/hero with which to identify;
- Proposing progressive levels of play that could follow the level of competence of the child;
- Using primary colors for designing the virtual environments, with bright tonality;
- Developing a videogame whose game session is no longer than 20/30 min;
- When designing the videogame, considering a price range of EUR 100–250 for the console and of EUR 40–60 for the videogame.

These indications are a useful starting point for the development of videogames that meet the actual market needs. Indeed, the recruited families highlighted the lack of inclusive, accessible, and enjoyable games and expressed their desire to approach this kind of videogame that could improve the quality of life of their children. Therefore, within the GiocAbile framework, we will develop an innovative videogame starting from the mentioned requirements, and we will evaluate it in terms of usability, acceptability, and user experience.

**Supplementary Materials:** The survey is available online at https://www.mdpi.com/article/10.3390/app11219886/s1.

**Author Contributions:** Conceptualization, E.B., M.P., O.P., M.M. (Matteo Malosio), S.P. and F.F.; Methodology, R.N., E.B., F.F., F.L., S.L.P., S.M., E.D., F.C., L.O., M.L.N., L.G., M.M. (Marta Mondellini), J.R., A.S. and V.A.; Formal Analysis, R.N.; Resources, M.V. and E.B.; Writing—Original Draft Preparation, R.N.; Writing—Review and Editing, R.N., E.B., M.P., O.P., M.M. (Matteo Malosio), M.M. (Marta Mondellini), S.P., F.F., F.L., S.L.P., L.G., V.A., M.V., J.R. and A.S.; Project Administration, M.V. and V.A. All authors have read and agreed to the published version of the manuscript.

**Funding:** This research was funded by Fondazione Cariplo (project CREW-Codesign for Rehabilitation and Wellbeing, GiocAbile concept, 2018/1471). The study was also partially supported by the Italian Ministry of Health (RC 2020/2021).

**Institutional Review Board Statement:** The study was conducted according to the guidelines of the Declaration of Helsinki and approved by the Ethics Committee of IRCCS Medea (GIP 399, 25/05/2017).

**Informed Consent Statement:** Patient consent was waived due to the anonymity of the survey and answers.

**Data Availability Statement:** The data presented in this study are openly available in Zenodo at 10.5281/zenodo.5457559 (link: https://zenodo.org/record/5457559#.YVXgXbUzZPY, accessed on 5 September 2021).

**Acknowledgments:** The authors thank all the families that participated to the survey.

**Conflicts of Interest:** The authors declare no conflict of interest. The funders had no role in the design of the study; in the collection, analyses, or interpretation of data; in the writing of the manuscript, or in the decision to publish the results.

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
