# Peer review of "What Children with Neuromotor Disabilities Need to Play with Technological Games"

_applsci, doi:10.3390/app11219886_

Round 1

Reviewer 1 Report

The work concerns the presence of games in the lives of handicapped children. This is undoubtedly an important topic that is not very often the subject of scientific research.

The research method adopted in this work is based on survey interviews with parents of children with neuromotor disabilities . The results of the research are recommendations for video game designers. Some of these recommendations and suggestions are quite obvious observations or well-known from other studies, but some of them may be useful. This applies to game design preferences, controllers and colors, gameplay duration, acceptable cost of the game product for parents.

The disadvantage of the research is too small a research sample. In addition, almost 40% of the respondents came from one region of Italy. Hence, most conclusions and recommendations require confirmation on a larger and more random sample.

Reviewer 2 Report

The article addresses timely, important topic: how we can make games accessible for children with disabilities. In this particular study, the results from a survey for the families of children with  neuromotor disabilities are presented along with design considerations for accessible games.

The study is well done, and reporting is detailed - even too much so, and the articlew would benefit from being more concise. The conclusions are clear, and it shows how further work in the area can be carried out.

This is an excellent article, but I have one major problem with it: the questionnaire is done for parents, not the children themselves. Even I can definitely understand the (pracrical) reasons for this, this is a major limitation of the work carried out. In particular, here almost half of the children prfer to play alone. As a researcher, I have seen how much the opinions of children differ from those of others (teachers, parents). As a parent, I could not expess the opinions of my children if asked questions like in this study. Therefore, in the future work the opinions of children should be emphasized, and the results here leave some doubts. I hope the authors will explain this shortcoming in the revised version of the article.

Reviewer 3 Report

This is a really timely and interesting study that explores some of the features needed to develop games, gaming platforms and accessories that are usable for children with physical impairments.  The work is motivated by an understanding that play is important for physical, cognitive and psychological development in all children.

I would like the authors to address a number of questions that arise from reading this version of the manuscript.

1. Introduction
Firstly, while the introduction is well written, the sudden mention of GiocAbile is not well motivated. Either leave out at this stage, or since your study is part of the GiocAbile project, provide more information to ground your work.  Also "fun" is a sloppy adjective to describe games - it could mean so many different things and no game designer would take it seriously.  Some more analysis of the extensive literature available on game design and player engagement would be appropriate.

I suggest starting with the idea that you undertook a study to find out about your users, and introduce the GiocAbile project in the discussion - pointing to future research that will be based on your findings. Then you might be able to offer some well-considered aims for the game you would like to design, as well as the list of technical indications.

2. Materials and Methods
Not sure why the technical software you used to create the survey is relevant. 

You state the survey was co-designed by "end-users" - which to me would imply the children, whereas in fact clinicians and charity president were your collaborative partners.  Since the point of the exercise was presumably to find out what the children themselves need and want, I'm not sure about your choice, so you should provide a strong reason for this.

3. Results
The supplementary material shows your survey questions, but it is not clear which were check-boxes (multiple checks possible) and which were "select-one" type questions.  The visual representations of data are very helpful.

Sections 1 and 2 seemed ok, but section 3 trivialises some very important game design aspects. For example, the game setting question (jungle, sky, seabed etc) seems pointless, as does the character question.  Asking about gameplay elements - types of game etc - was much more relevant.

I thought the range of controllers was fascinating and should be explained more.  My issue is how you expected parents to make a choice on behalf of their children who might never have seen or used some of these controllers before.  The results suggest that people mainly chose items they were familiar with, but you don't address this. This section is potentially very interesting and definitely needs more work.

4. Discussion
Here you introduce GiocAbile and offer a detailed discussion of many relevant choices made by parents.  I would like to know if the children were involved in the survey in any way, and your thoughts regarding how you might be able to include them in the design process in the future.  

Good luck with the project!
